# Automatic Speech Recognition (ASR) Systems Applied to Pronunciation Assessment of L2 Spanish for Japanese Speakers [†]

Cristian Tejedor-García [1,2,*,‡], Valentín Cardeñoso-Payo [2,*,‡] and David Escudero-Mancebo [2,*,‡]

1   Centre for Language and Speech Technology (CLST), Radboud University Nijmegen, P.O. Box 9103, 6500 Nijmegen, The Netherlands
2   ECA-SIMM Research Group, Department of Computer Science, University of Valladolid, 47002 Valladolid, Spain
*   Correspondence: cristian.tejedorgarcia@ru.nl (C.T.-G.); valen@infor.uva.es (V.C.-P.); descuder@infor.uva.es (D.E.-M.)
†   This paper is an extended version of our paper published in the conference IberSPEECH2020.
‡   These authors contributed equally to this work.

**Featured Application: The CAPT tool, ASR technology and procedure described in this work can be successfully applied to support typical learning paces for Spanish as a foreign language for Japanese people. With small changes, the application can be tailored to a different target L2, if the set of minimal pairs used for the discrimination, pronunciation and mixed-mode activities is adapted to the specific L1–L2 pair.**

**Abstract:** General-purpose automatic speech recognition (ASR) systems have improved in quality and are being used for pronunciation assessment. However, the assessment of isolated short utterances, such as words in minimal pairs for segmental approaches, remains an important challenge, even more so for non-native speakers. In this work, we compare the performance of our own tailored ASR system (kASR) with the one of Google ASR (gASR) for the assessment of Spanish minimal pair words produced by 33 native Japanese speakers in a computer-assisted pronunciation training (CAPT) scenario. Participants in a pre/post-test training experiment spanning four weeks were split into three groups: experimental, in-classroom, and placebo. The experimental group used the CAPT tool described in the paper, which we specially designed for autonomous pronunciation training. A statistically significant improvement for the experimental and in-classroom groups was revealed, and moderate correlation values between gASR and kASR results were obtained, in addition to strong correlations between the post-test scores of both ASR systems and the CAPT application scores found at the final stages of application use. These results suggest that both ASR alternatives are valid for assessing minimal pairs in CAPT tools, in the current configuration. Discussion on possible ways to improve our system and possibilities for future research are included.

**Keywords:** automatic speech recognition (ASR); automatic assessment tools; foreign language pronunciation; pronunciation training; computer-assisted pronunciation training (CAPT); automatic pronunciation assessment; learning environments; minimal pairs

## 1. Introduction

Recent advances in automatic speech recognition (ASR) have made this technology a potential solution for transcribing audio input for computer-assisted pronunciation training (CAPT) tools [1,2]. Available ASR technology, properly adapted, might help human instructors with pronunciation assessment tasks, freeing them from hours of tedious work, allowing for the simultaneous and fast assessment of several students, and providing a form of assessment that is not affected by subjectivity, emotion, fatigue, or accidental

lack of concentration [3]. Thus, ASR systems can help in the assessment and feedback of learner production, reducing human costs [4,5]. Although most of the scarce empirical studies which include ASR technology in CAPT tools assess sentences in large portions of either reading or spontaneous speech [6,7], the assessment of words in isolation remains a substantial challenge [8,9].

General-purpose off-the-shelf ASR systems such as Google ASR (https://cloud.google.com/speech-to-text, accessed on 27 June 2021) (gASR) are becoming progressively more popular each day due to their easy accessibility, scalability, and, most importantly, effectiveness [10,11]. These services provide accurate speech-to-text capabilities to companies and academics who might not have the possibility of training, developing, and maintaining a specific-purpose ASR system. However, despite the advantages of these systems (e.g., they are trained on large datasets and span different domains), there is an obvious need for improving their performance when used on in-domain data-specific scenarios, such as segmental approaches in CAPT for non-native speakers. Concerning the existing ASR toolkits, Kaldi has shown its leading role in recent years, with its advantages of having flexible and modern code that is easy to understand, modify, and extend [12], becoming a highly matured development tool for almost any language [13,14].

English is the most frequently addressed L2 in CAPT experiments [6] and in commercial language learning applications, such as Duolingo (https://www.duolingo.com/, accessed on 27 June 2021) or Babbel (https://www.babbel.com/, accessed on 27 June 2021). However, there are scarce empirical experiments in the state-of-the-art which focus on pronunciation instruction and assessment for native Japanese learners of Spanish as a foreign language, and as far as we are concerned, no one has included ASR technology. For instance, 1440 utterances of Japanese learners of Spanish as a foreign language (A1–A2) were analyzed manually with Praat by phonetics experts in [15]. Students performed different perception and production tasks with an instructor, and they achieved positive significant differences (at the segmental level) between the pre-test and post-test values. A pilot study on the perception of Spanish stress by Japanese learners of Spanish was reported in [16]. Native and non-native participants listened to natural speech recorded by a native Spanish speaker and were asked to mark one of three possibilities (the same word with three stress variants) on an answer sheet. Non-native speech was manually transcribed with Praat by phonetic experts in [17], in an attempt to establish rule-based strategies for labeling intermediate realizations, helping to detect both canonical and erroneous realizations in a potential error detection system. Different perception tasks were carried out in [18]. It was reported how the speakers of native language (L1) Japanese tend to perceive Spanish /y/ when it is pronounced by native speakers of Spanish, and how the L1 Spanish and L1 Japanese listeners evaluate and accept various consonants as allophones of Spanish /y/, comparing both groups.

In previous work, we presented the development and the first pilot test of a CAPT application with ASR and text-to-speech (TTS) technology, Japañol, through a training protocol [19,20]. This learning application for smart devices includes a specific exposure–perception–production cycle of training activities with minimal pairs, which are presented to students in lessons on the most difficult Spanish constructs for native Japanese speakers. We were able to empirically measure a statistically significant improvement between the pre and post-test values of eight native Japanese speakers in a single experimental group. The students' utterances were assessed by experts in phonetics and by the gASR system, obtaining strong correlations between human and machine values. After this first pilot test, we wanted to take a step further and to find pronunciation mistakes associated with key features of the proficiency-level characterization of more participants (33) and different groups (3). However, assessing such a quantity of utterances by human raters can lead to problems regarding time and resources. Furthermore, the gASR pricing policy and its limited black-box functionalities also motivated us to look for alternatives to assess all the utterances, developing a specific ASR system for Spanish from scratch, using Kaldi (kASR). In this work, we analyze the audio utterances of the pre-test and post-test of 33

Japanese learners of Spanish as a foreign language with two different ASR systems (gASR and kASR) to address the research question of how these general and specific-purpose ASR systems compare in the assessment of short isolated words used as challenges in a learning application for CAPT.

This paper is organized as follows. The experimental procedure is described in Section 2, which includes the participants and protocol definition, a description of the CAPT tool, a brief description of the process for elaborating the kASR system, and the collection of metrics and instruments for collecting the necessary data. The Results section presents, on one hand, the results of the training of the users that worked with the CAPT tool and, on the other hand, the performance of the two versions of the ASR systems: the word error rate (WER) values of the kASR system developed, the pronunciation assessment of the participants at the beginning and at the end of the experiment, including intra- and inter-group differences, and the ASR scores' correlation of both ASR systems. Then, we discuss the user interaction with the CAPT tool, the performance of both state-of-the-art ASR systems in CAPT, and we shed light on lines of future work. Finally, we end this paper with the main conclusions.

## 2. Experimental Procedure

Figure 1 shows the experimental procedure followed in this work. At the bottom, we see that a set of recordings of native speakers is used to train a kASR of Spanish words. On the upper part of the diagram, we see that a group of non-native speakers are evaluated in pre/post-tests in order to measure improvements after training. Speakers are separated into three different groups (placebo, in-classroom, and experimental) to compare different conditions. Both the utterances of the pre/post-tests and the interactions with the software tool (experimental group) are recorded, so that a corpus of non-native speech is collected. The non-native audio files are then evaluated with both the gASR and the kASR systems, so that the students' performance during training can be analyzed.

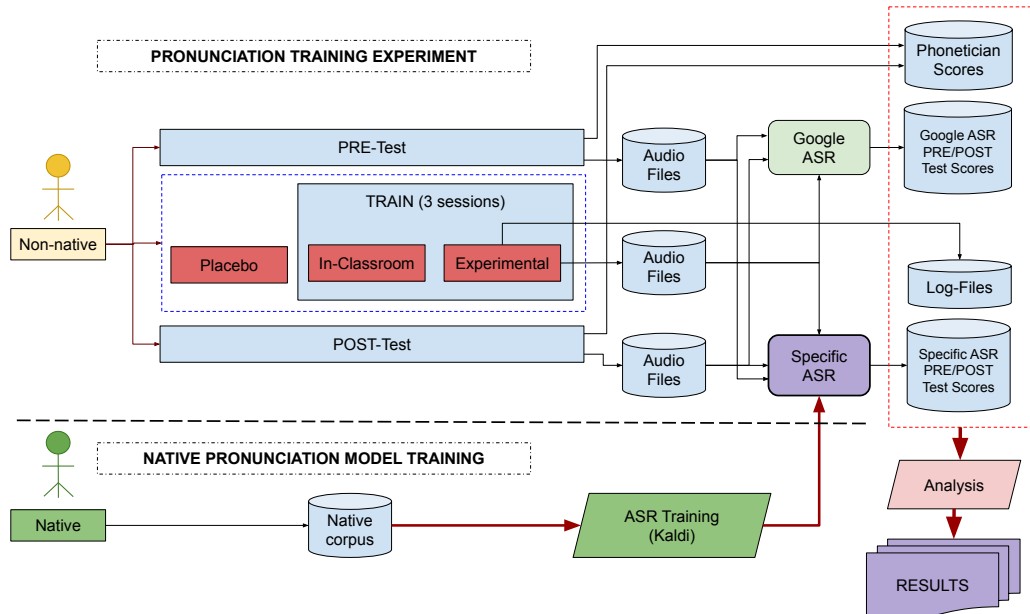

**Figure 1.** Scheme of the experimental procedure.

The whole procedure can be compared with the one used in previous experiments [19,20] where human-based scores (provided by expert phoneticians) were used. Section 2.1 describes the set of informants that participated in the evaluation and audio recordings. Section 2.2 describes the protocol of the training sessions, including details of the pre- and post-tests. Section 2.5 shows the training of the kASR system. Section 2.6 presents the instruments and metrics used for the evaluation of the experiment.

## 2.1. Participants

A total of 33 native Japanese speakers aged between 18 and 26 years participated voluntarily in the evaluation of the experimental prototype. Participants came from two different locations: 8 students (5 female, 3 male) were registered in a Spanish intensive course provided at the Language Center of the University of Valladolid and had recently arrived in Spain from Japan in order to start the L2 Spanish course; the remainder comprised 25 female students of the Spanish philology degree from the University of Seisen, Japan. The results of the first location (Valladolid) allowed us to verify that there were no particularly differentiating aspects in the results analyzed by gender [19]. Therefore, we did not expect the fact that all participants were female in the location to have a significant impact on the results. All of them declared a low level of Spanish as a foreign language, with no previous training in Spanish phonetics. None of them had stayed in any Spanish speaking country for more than 3 months. Furthermore, they were requested not to complete any extra work in Spanish (e.g., conversation exchanges with natives or extra phonetics research) while the experiment was still active.

Participants were randomly divided into three groups: (1) **experimental group,** 18 students (15 female, 3 male) who trained their Spanish pronunciation with Japañol, during three sessions of 60 min; (2) **in-classroom group**, 8 female students who attended three 60 min pronunciation teaching sessions within the Spanish course, with their usual instructor, making no use of any computer-assisted interactive tools; and (3) **placebo group**, 7 female students who only took the pre-test and post-test. They attended neither the classroom nor the laboratory for Spanish phonetics instruction.

## 2.2. Protocol Description

We followed a four-week protocol which included a pre-test, three training sessions, and a post-test for the non-native participants (see Appendix A to see the content of the tests). Native speakers recorded the speech training corpus for the kASR system. At the beginning, the non-native subjects took part in the pre-test session individually in a quiet testing room. The utterances were recorded with a microphone and an audio recorder (the procedure was the same for the post-test). All the students took the pre-test under the sole supervision of a member of the research team. They were asked to read aloud the 28 minimal pairs administered via a sheet of paper with no time limitation (https://github.com/eca-simm/minimal-pairs-japanol-eses-jpjp, accessed on 27 June 2021). The pairs came from 7 contrasts containing Spanish consonant sounds considered the most difficult to perceive and produce by native Japanese speakers (see more details in [19]): [θ]–[f], [θ]–[s], [fu]–[xu], [l]–[ɾ], [l]–[r], [ɾ]–[rr], and [fl]–[fɾ]. Students were free to repeat each contrast as many times as they wished if they thought they might have mispronounced them.

From the same 7 contrasts, a total of 84 minimal pairs (https://github.com/eca-simm/minimal-pairs-japanol-eses-jpjp, accessed on 27 June 2021) were presented to the experimental and in-classroom group participants in 7 lessons across three training sessions. The minimal pairs were carefully selected by experts considering the gASR limitations (homophones, word-frequency, very short words, and out-of-context words, in a similar process as in [8]). The lessons were included in the CAPT tool for the experimental group and during the class sessions for the in-classroom group (12 minimal pairs per lesson, 2 lessons per session, except for the last session that included 3 lessons; see more details about the training activities in [19]). The training protocol sessions were carried out during students' course lectures in the classroom, in which a minimal pair was practiced in each lesson (blocked practice) and most phonemes were practiced again in later sessions (spaced practice). Regarding the sounds practiced in each session, in the first one, sounds [fu]–[xu] and [l]–[ɾ] were contrasted, then [l]–[r] and [ɾ]–[rr], and the last session involved the sounds [fl]–[fɾ], [θ]–[f], and [θ]–[s]. Finally, subjects of the placebo group did not participate in the training sessions. They were supposed to take the pre-test and post-test and obtain

results without significant differences. All participants were awarded with a diploma and a reward after completing all stages of the experiment.

*2.3. Description of the CAPT Mobile Application*

To carry out all the experiments, we built a mobile app, Japañol, starting from a previous prototype app designed for self-directed training of English as an L2 [8]. Figure 2 shows the regular sequence of steps to complete a lesson in Japañol. After user authentication in (step 1), seven lessons are presented at the main menu of the application (step 2). Each lesson includes a pair of Spanish sound contrasts and users achieve a particular score, expressed as a percentage. Lessons are divided into five main training modes, Theory, Exposure, Discrimination, Pronunciation, and Mixed Modes (step 3), in which each one proposes several task types with a fixed number of mandatory task tokens. The final lesson score is the mean score of the last three modes. Users are guided by the system in order to complete all training modes of a lesson. When reaching a score below 60% in Discrimination, Pronunciation, or Mixed Modes, users are recommended to return to Exposure mode as a feedback resource and then return to the failed mode. Moreover, the next lesson is enabled when users reach a minimum score of 60%.

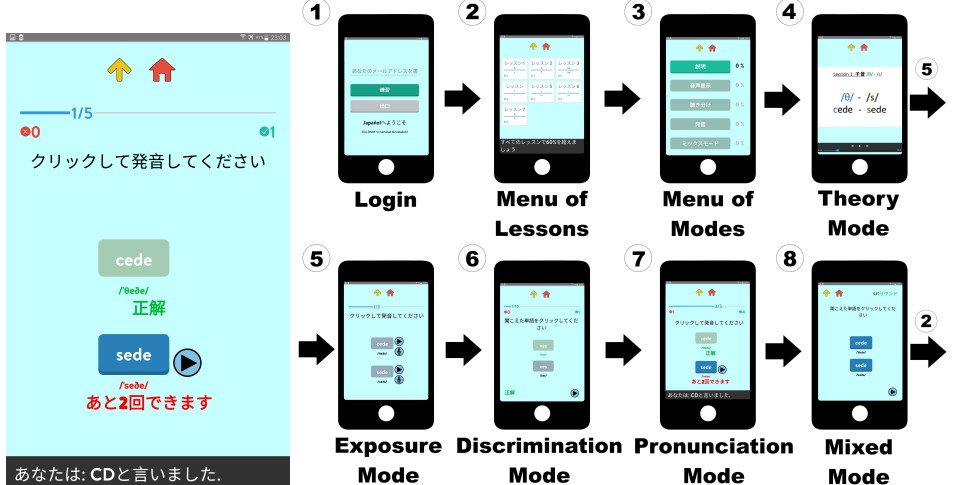

**Figure 2.** Standard flow to complete a lesson in Japañol.

The first training mode is **Theory** (step 4). A brief and simple video describing the target contrast of the lesson is presented to the user as the first contact with feedback. At the end of the video, the next mode becomes available, but users may choose to review the material as many times as they want. **Exposure** (step 5) is the second mode. Users strengthen the lesson contrast experience previously introduced in Theory mode, in order to support their assimilation. Three minimal pairs are displayed to the user. In each one of them, both words are synthetically produced by *Google* TTS five times (highlighting the current word), alternately and slowly. After this, users must record themselves at least one time per word and listen to their own and the system's sound. Words are represented with their orthographic and phonemic forms. A replay button allows users to listen to the specified word again. Synthetic output is produced by *Google*'s offline Text-To-Speech tool for Android. After all previous required events per minimal pair (listen-record-compare), participants are allowed to remain in this mode for as long as they wish, listening, recording, and comparing at will, before returning to the Modes menu. Step 6 refers to **Discrimination** mode, in which ten minimal pairs are presented to the user consecutively. In each one of them, one of the words is synthetically produced, randomly. The challenge of this mode consists of identifying which word is produced. As feedback elements, words have their orthographic and phonetic transcription representations. Users can also request to listen to the target word again with a replay button. Speed varies alternately between slow and normal speed rates. Finally, the system changes the word color to green (success)

or red (failure) with a chime sound. **Pronunciation** is the fourth mode (step 7), whose aim is to produce, as well as possible, both words, separately, of the five minimal pairs presented with their phonetic transcription. gASR determines automatically and in real time acceptable or non-acceptable inputs. In each production attempt, the tool displays a text message with the recognized speech, plays a right/wrong sound, and changes the word's color to green or red. The maximum number of attempts per word is five in order not to discourage users. However, after three consecutive failures, the system offers to the user the possibility of requesting a word synthesis as an explicit feedback as many times as they want with a replay button. **Mixed mode** is the last mode of each lesson (step 8). Nine production and perception tasks alternate at random in order to further consolidate the obtained skills and knowledge. Regarding listening tasks with the TTS, mandatory listenings are those which are associated with mandatory activities with the tool and non-mandatory listenings are those which are freely undertaken by the user whenever she has doubts about the pronunciation of a given word.

### 2.4. Native Corpus Preparation

A group of 10 native Spanish speakers from the theater company Pie Izquierdo of Valladolid (5 women and 5 men) participated in the recording of a total of 41,000 utterances (7.1 h of speech data) for the training corpus of the kASR system for assessing the students' utterances gathered during the experimentation.

Each one of the native speakers recorded individually 164 words (https://github.com/eca-simm/minimal-pairs-japanol-eses-jpjp, accessed on 27 June 2021) 25 times, (41,000 utterances in total) presented randomly in five-hour sessions, for elaborating the training corpus for the kASR system. The average, minimum, maximum, and standard deviation of the word lengths were: 4.29, 2, 8, and 1.07, respectively. The phoneme frequency (%) was: [a]: 16.9, [o]: 11.3, [r]: 9.0, [e]: 7.8, [f]: 5.3, [s]: 5.0, [ɾ]: 4.8, [l]: 4.5, [t]: 3.6, [k]: 3.6, [u]: 3.2, [i]: 3.2, [θ]: 3.2, [n]: 2.8, [m]: 2.3, [ɣ]: 1.8, [j]: 1.4, [ð]: 1.5, [x]: 1.3, [b]: 1.3, [p]: 1.1, [d]: 1.1, [β]: 0.9, [w]: 0.9, [ŋ]: 0.7, [g]: 0.3, [ʤ]: 0.2, and [z]: 0.1.

The recording sessions were carried out in an anechoic chamber at the University of Valladolid with the help of a member of the ECA-SIMM research group. The machine configuration on which the kASR system was installed was CentOS 8 (64-bit operating system), Intel(R) Core(TM) i7-8700K CPU (12 cores) processor with 3.70 GHz.

### 2.5. Developing an ASR System with Kaldi

Until now, in our previous works, we have always used gASR for the automatic assessment of pre/post-tests as a complement to or replacement for subjective assessment. Since gASR works as a black-box and does not allow us to obtain details on the quality of each individual speech fragment, we decided to develop an in-house ASR system of our own using Kaldi (kASR). In this subsection, we present the ASR pipeline that we implemented for the kASR system and we provide details about its general architecture and specific parameters.

Our kASR system uses a standard context-dependent triphone system with a simple Gaussian Mixture Model–Hidden Markov Model (GMM-–HMM) [21], adapted from existing Kaldi recipes [22]. Although recent studies report excellent outcomes from neural models with Kaldi [23], we did not find relevant differences in preliminary runs due to, mainly, the small size of the training corpus (described in Section 2.4). After collecting and preparing the speech data for training and testing, the first step is to extract acoustic features from the audio utterances and training monophone models. These features are Mel frequency cepstral coefficients (MFCCs) with per-speaker cepstral mean and variance statistics. Since Kaldi is based on a finite-state transducer-based framework to build language models from the raw text, we use the SRILM toolkit for building a 2-g language model [24].

To train a model, monophone GMMs are first iteratively trained and used to generate a basic alignment. Triphone GMMs are then trained to take the surrounding phonetic

context into account, in addition to clustering of triphones to combat sparsity. The triphone models are used to generate alignments, which are then used for learning acoustic feature transforms on a per-speaker basis in order to make them more suited to speakers in other datasets [25]. In our case, we set 2000 total Gaussian components for the monophone training. Then, we realigned and retrained these models four times (tri4) with 5 states per HMM. In particular, in the first triphone pass, we used MFCCs, delta, and delta–delta features (2500 leaves and 15,000 Gaussian components); in the second triphone pass, we included linear discriminant analysis (LDA) and Maximum Likelihood Linear Transform (MLLT) with 3500 leaves and 20,000 Gaussian components; the third triphone pass combined LDA and MLLT with 4200 leaves and 40,000 Gaussian components, and the final step (tri4) included LDA, MLLT, and speaker adaptive training (SAT) with 5000 leaves and 50,000 Gaussian components. The language model was a bigram with 164 unique words (same probability) for the lexicon, 26 nonsilence phones, and the standard SIL and UNK phones.

### 2.6. Instruments and Metrics

We gathered data from five different sources: (1) a registration form with students' demographic information, (2) pre-test utterances, (3) log files, (4) utterances of users' interactions with Japañol, and (5) post-test utterances. Personal information included name, age, gender, L1, academic level, and final consent to analyze all gathered data. Log files gathered all low-level interaction events with the CAPT tool and monitored all user activities with timestamps. From these files, we computed a CAPT score per speaker which refers to the final performance at the end of the experiment. It includes the number of correct answers in both perception and production (in which we used gASR) tasks while training with Japañol [19]. Pre/post-test utterances consisted in oral productions of the minimal pairs lists provided to the students.

A set of experimental variables was computed: (1) WER values of the train/test set models for the specific-purpose kASR system developed in a [0, 100] scale; (2) the student's pronunciation improvement at the segmental level comparing the difference between the number of correct words at the beginning (pre-test) and at the end (post-test) of the experiment in a [0, 10] scale. We used this scale for helping teachers to understand the score as they use it in the course's exams. This value consists of the mean of correct productions in relation to the total number of utterances. Finally, we used (3) the correlation values between gASR and kASR systems of the pre/post-test utterances and between the CAPT score and both ASR systems at the end of the experiment (post-test) in a [0, 1] scale.

By way of statistical metrics and indexes, Wilcoxon signed-rank tests were used to compare the differences between the pre/post-test utterances of each group (intra-group), Mann–Whitney U tests were used to compare the differences between the groups (inter-group), and Pearson correlations were used to explain the statistical relationship between the values of the ASR systems and the final CAPT scores.

## 3. Results

### 3.1. User Interaction with the CAPT Tool

Table 1 displays the results related to the user interaction with the CAPT system (experimental group, 18 participants). Columns $\bar{n}$, $m$, and $M$ are the mean, minimum, and maximum values, respectively. *Time (min)* row stands for the time spent (minutes) per learner in each training mode in the three sessions of the experiment. *#Tries* represents the number of times a mode was executed by each user. The symbol — stands for 'not applicable'. *Mand.* and *Req.* mean mandatory and requested listenings (see Section 2.3). The TTS system was used in both listening types, whereas the ASR was only used in the *#Productions* row.

**Table 1.** User's training activities with the CAPT system.

| | Theory | | | Exposure | | | Discrimination | | | Pronunciation | | | Mixed | | |
|---|---|---|---|---|---|---|---|---|---|---|---|---|---|---|---|
| | $\bar{n}$ | *m* | *M* | $\bar{n}$ | *m* | *M* | $\bar{n}$ | *m* | *M* | $\bar{n}$ | *m* | *M* | $\bar{n}$ | *m* | *M* |
| Time (min) | 14.80 | 8.7 | 20.8 | 19.7 | 12.8 | 21.9 | 7.1 | 4.1 | 13.8 | 43.6 | 22.4 | 72.9 | 17.0 | 7.6 | 30.5 |
| #Tries | 7.8 | 6 | 10 | 10.6 | 7 | 16 | 8.5 | 7 | 15 | 10.1 | 7 | 14 | 6.7 | 3 | 10 |
| #Mand.List. | - | - | - | 287.8 | 210 | 390 | 91.7 | 70 | 134 | - | - | - | 20.2 | 9 | 30 |
| #Req.List. | - | - | - | 99.3 | 53 | 157 | 33.0 | 0 | 153 | 54.9 | 0 | 127 | 25.6 | 6 | 60 |
| #Discriminations | - | - | - | - | - | - | 91.7 | 70 | 134 | - | - | - | 20.2 | 9 | 30 |
| #Productions | - | - | - | - | - | - | - | - | - | 208.8 | 116 | 356 | 82.9 | 38 | 181 |
| #Recordings | - | - | - | 62.4 | 42 | 81 | - | - | - | - | - | - | - | - | - |

Table 1 shows that there are important differences in the level of use of the tool depending on the user. For instance, the fastest learner performing pronunciation activities spent 22.43 min, whereas the slowest one took 72.85 min. This contrast can also be observed in the time spent on the rest of the training modes and in the number of times that the learners practiced each one of them (row *#Tries*). Overall, 85.25% of the time was consumed by carrying out interactive training modes (Exposure, Discrimination, Pronunciation, and Mixed Modes). The inter-user differences affected both the number of times the users made use of the ASR (154 minimum vs. 537 maximum) and the number of times they requested the use of TTS (59 vs. 497 times), reaching a rate of 9.0 uses of the speech technologies per minute.

Tables 2 and 3 show the confusion matrices between the sounds of the minimal pairs in perception and production events, since the sounds were presented in pairs in each lesson. In both tables, the rows are the phonemes expected by the tool and the columns are the phonemes selected (discrimination training mode) or produced (production training mod) by the user. These produced phonemes are derived from the word recognized by the gASR, not because we look directly at the phoneme recognized, since gASR does not provide us with phoneme-level segmentation. *TPR* is the true positive rate or recall. The symbol—stands for 'not applicable'. *#Lis* is the number of requested (e.g., non-mandatory) listenings of the word in the minimal pair including the sound of the phoneme in each row.

**Table 2.** Confusion matrix of discrimination tasks (diagonal: right discrimination tasks).

| | | | | | | Discrimination Tasks | | | | | |
|---|---|---|---|---|---|---|---|---|---|---|---|
| *#Lis* | | [fl] | [fɾ] | [l] | [ɾ] | [rr] | [s] | [θ] | [f] | [fu] | [xu] | *TPR* (%) |
| 65 | [fl] | *123* | 64 | - | - | - | - | - | - | - | - | 65.8% |
| 52 | [fɾ] | 69 | *115* | - | - | - | - | - | - | - | - | 62.5% |
| 139 | [l] | - | - | *239* | 56 | 19 | - | - | - | - | - | 76.1% |
| 115 | [ɾ] | - | - | 71 | *217* | 16 | - | - | - | - | - | 71.4% |
| 51 | [rr] | - | - | 15 | 21 | *215* | - | - | - | - | - | 85.7% |
| 45 | [s] | - | - | - | - | - | *95* | 32 | - | - | - | 74.8% |
| 45 | [θ] | - | - | - | - | - | 15 | *214* | 11 | - | - | 89.2% |
| 16 | [f] | - | - | - | - | - | - | 4 | *104* | - | - | 96.3% |
| 89 | [fu] | - | - | - | - | - | - | - | - | *115* | 34 | 77.2% |
| 103 | [xu] | - | - | - | - | - | - | - | - | 39 | *111* | 74.0% |

As shown in Tables 2 and 3, the most confused pairs in discrimination tasks were [l]–[ɾ], both individually (56 and 71, 127 times) and preceded by the sound [f] (69 and 64, 133 times). Furthermore, the number of requested listenings related to these sounds was the highest one (65 and 139, 204 times for [l] and 167 (52 and 115) for [ɾ]). The least confused pair in discrimination was [θ]–[f] (11 and 4, 15 times). The sounds with the lowest discrimination *TPR* rate were [fl] and [fɾ] (both < 66.0%), and those with the highest discrimination *TPR* rate were [θ] and [f] (both > 89%), corresponding also to the lowest number of requested listenings (45 and 16, respectively).

Table 3 shows the results related to production events per word utterance. *#Lis* is the number of requested listenings of the corresponding *sound* row at (*first* | *last*) attempt.

A positive improvement from first to last attempt was observed (*TPR* column), with the highest ones being the [fl] (33.2%) and [fɾ] (21.1%) sounds. In particular, these two sounds constituted the most confused pair in first-attempt production tasks (73 and 79, 152 times), where the least confused one was [l]–[rr] (37 and 22, 59 times). The sounds with the lowest production *TPR* rate were [fl] and [s] (both < 47%), and those with the highest production *TPR* rates were [ɾ] and [rr] (both > 73%). On the other hand, the most confused pair in last-attempt production tasks was [fu]–[xu] (91 and 106, 197 times), reaching the lowest production *TPR* rates (56.6% and 60.6%, respectively). Moreover, the number of requested listenings was the highest in both cases (240 and 186, respectively). The least confused pair was [l]–[rr] (9 and 14, 23 times), reaching *TPR* rate values higher than 85%.

**Table 3.** Confusion matrix of production tasks at first and last attempt per word sequence (diagonal: right production tasks at first and last attempt per word sequence).

| | | **Production Tasks (First Attempt\|Last Attempt)** | | | | | | | | | | |
|---|---|---|---|---|---|---|---|---|---|---|---|---|
| *#Lis* | | **[fl]** | **[fɾ]** | **[l]** | **[ɾ]** | **[rr]** | **[s]** | **[θ]** | **[f]** | **[fu]** | **[xu]** | *TPR* **(%)** |
| 13\|128 | [fl] | *65\|170* | 79\|47 | - | - | - | - | - | - | - | - | 45.1%\|78.3% |
| 3\|125 | [fɾ] | 73\|64 | *65\|137* | - | - | - | - | - | - | - | - | 47.1%\|68.2% |
| 9\|105 | [l] | - | - | *177\|253* | 45\|31 | 37\|14 | - | - | - | - | - | 68.3%\|84.9% |
| 8\|103 | [ɾ] | - | - | 33\|21 | *209\|289* | 42\|14 | - | - | - | - | - | 73.6%\|89.2% |
| 3\|70 | [rr] | - | - | 22\|9 | 44\|22 | *189\|252* | - | - | - | - | - | 74.1%\|89.0% |
| 6\|146 | [s] | - | - | - | - | - | *58\|134* | 66\|67 | - | - | - | 46.8%\|66.7% |
| 2\|202 | [θ] | - | - | - | - | - | 79\|96 | *142\|226* | 38\|12 | - | - | 54.8%\|67.7% |
| 0\|29 | [f] | - | - | - | - | - | - | 38\|19 | *97\|116* | - | - | 71.9%\|85.9% |
| 4\|240 | [fu] | - | - | - | - | - | - | - | - | *62\|138* | 62\|106 | 50.0%\|56.6% |
| 5\|186 | [xu] | - | - | - | - | - | - | - | - | 59\|91 | *63\|140* | 51.6%\|60.6% |

### 3.2. ASR Performance

We tested the speech utterances of the pre/post-tests with two different ASR systems, i.e., the general-purpose gASR and a specific-purpose ASR created from scratch with Kaldi (kASR), to validate that the results using kASR were not casual. We considered a comparison with other Spanish ASR systems [26] not to be informative or fair to carry out, since our kASR system is not a general-purpose one, but a tailored ASR with a closed set of words related to minimal pairs. Table 4 shows the WER values obtained by both ASR systems used in the experimentation with two different sources of speech data (native and non-native).

**Table 4.** WER values (%) of the experiment's ASR systems.

| | **Train Model** | | | | | | |
|---|---|---|---|---|---|---|---|
| | **gASR** | **kASR** | | | | | |
| | | *All* | *Female* | *Male* | *Best1* | *Best2* | *Best3* |
| *Native* | 5.0 | 0.0024 | 3.10 | 1.55 | 0.14 | 0.14 | 0.23 |
| *Non-native* | 30.0 | 44.22 | 55.91 | 64.12 | 46.40 | 46.98 | 48.08 |

Regarding the native models (Table 4), the *All* model included 41,000 utterances of the native speakers in the train set. The *Female* model included 20,500 utterances of the five female native speakers in the train set. The *Male* model included 20,500 utterances of the five male native speakers in the train set. The *Best1*, *Best2*, and *Best3* models included 32,800 utterances (80%) of the total native speakers (4 females and 4 males) in the train set. These last three models were obtained by comparing the WER values of all possible 80%/20% combinations (train/test sets) of the native speakers (e.g., 4 female and 4 male native speakers for training: 80%, and 1 female and 1 male for testing: 20%), and choosing the best three WER values (the lowest ones). On the other hand, the non-native test model

consisted of 3696 utterances (33 participants × 28 minimal pairs × 2 words per minimal pair × 2 tests).

The 5.0% WER value reported by Google for their English ASR system for native speech [10] corresponds to our WER value for native speech data. Google training techniques are applied also for their ASR in other majority languages, such as Spanish. Regarding our kASR system, we achieved values lower than 5.0% for native speech for the specific battery of minimal pairs introduced in Section 2 (e.g., *All* model: 0.0024%). On the other hand, we tested the non-native minimal pairs utterances with gASR, obtaining a 30.0% WER (16.0% non-recognized words). In the case of the kASR, as expected, the *All* model reported the best test results (44.22%) for the non-native speech. The *Female* train model yielded a better WER value for the non-native test model (55.91%) than the *Male* one (64.12%) since 30 out of 33 participants were female speakers.

Table 5 displays the average scores assigned by the gASR and kASR systems to the 3696 utterances of the pre/post-tests classified by the three groups of participants, in a [0, 10] scale. Symbols $\bar{n}$, N, and Δ refer to the mean score of the correct pre/post-test utterances, the number of utterances, and the difference between the post-test and pre-test average scores, respectively. The students who trained with the tool (experimental group) achieved the best pronunciation improvement values in both gASR (0.7) and kASR (1.1) systems. Nevertheless, the in-classroom group achieved better results in both tests and with both ASR systems (4.1 and 6.1 in the post-test; and 3.5 and 5.2 in the pre-test, gASR and kASR, respectively). The placebo group achieved the worst post-test values (3.2 and 3.5) and pronunciation improvement (Δ) values (0.2 and 0.4).

**Table 5.** Pre/post-test scores assigned by both ASR systems.

| Group | Pre-Test gASR $\bar{n}$ | Pre-Test gASR N | Pre-Test kASR $\bar{n}$ | Pre-Test kASR N | Post-Test gASR $\bar{n}$ | Post-Test gASR N | Post-Test kASR $\bar{n}$ | Post-Test kASR N | Δ (Post-Test–Pre-Test) gASR Δ | Δ (Post-Test–Pre-Test) kASR Δ |
|---|---|---|---|---|---|---|---|---|---|---|
| Experimental | 3.0 | 560 | 4.1 | 560 | 3.7 | 560 | 5.2 | 560 | 0.7 | 1.1 |
| In-classroom | 3.5 | 448 | 5.2 | 448 | 4.1 | 448 | 6.1 | 448 | 0.6 | 0.9 |
| Placebo | 3.0 | 392 | 3.1 | 392 | 3.2 | 392 | 3.5 | 392 | 0.2 | 0.4 |

Table 6 shows the cases in which there are statistically significant inter- and intra-group differences between the pre/post-test values. A Mann–Whitney U test found statistically significant differences between all groups and with both ASR systems in the post-test. Although there were significant differences between the pre-test scores of the in-classroom group and the experimental group, and the placebo group, such differences were minimal since the effect size values were small (r = 0.10 and r = 0.20, respectively). Regarding intra-group differences, a Wilcoxon signed-rank test (right part of Table 6) found statistically significant differences between the pre/post-test values of the experimental and in-classroom groups with both ASR systems. In the case of the placebo group, there were differences only in the gASR values.

**Table 6.** Inter and intra-group statistically significant differences between the scores of the pre/post-tests.

| Groups | Inter-Group (Mann–Whitney U) Pre-Test gASR p-Value | Pre-Test gASR Z | Pre-Test kASR p-Value | Pre-Test kASR Z | Post-Test gASR p-Value | Post-Test gASR Z | Post-Test kASR p-Value | Post-Test kASR Z | Intra-Group (Wilcoxon Signed-Rank) Post-Test–Pre-Test Group | gASR p-Value | gASR Z | kASR p-Value | kASR Z |
|---|---|---|---|---|---|---|---|---|---|---|---|---|---|
| EXP-INC | <0.001 | −8.892 | <0.001 | −3.645 | <0.001 | −2.773 | <0.001 | −2.886 | EXP | <0.001 | −13.784 | <0.001 | −5.448 |
| EXP-PLA | - | - | - | - | <0.001 | −5.324 | <0.001 | −3.527 | INC | <0.001 | −2.888 | <0.001 | −3.992 |
| INC-PLA | <0.001 | −8.050 | <0.001 | −3.431 | <0.001 | −6.32 | <0.001 | −7.651 | PLA | 0.002 | −3.154 | - | - |

EXP = Experimental group; INC = In-classroom group; PLA = Placebo group; - = No differences.

This learning difference at the end of the experiment was supported by the time spent by the speakers on carrying out the pre-test and post-test. Each participant took an average

of 83.77 s to complete the pre-test (63.85 s min. and 129 s max.) and an average of 94.10 s to complete the post-test (52.45 and 138.87 s min. and max.).

Finally, we analyzed the correlations between (1) the pre/post-test scores of both ASR systems (three groups) and (2) the Japañol CAPT tool scores with the experimental group's post-test scores of both ASR systems (since the experimental group was the only group with a CAPT score) in order to compare the three sources of objective scoring (Table 7).

**Table 7.** Regression coefficients of the CAPT, gASR, and kASR systems.

| *x* | *y* | *a* | *b* | *S.E.* | *r* | *p*-Value |
|---|---|---|---|---|---|---|
| pre-kASR | pre-gASR | 0.927 | 1.919 | 0.333 | 0.51 | 0.005 |
| post-kASR | post-gASR | 0.934 | 1.897 | 0.283 | 0.57 | 0.002 |
| post-gASR | CAPT score | 0.575 | −0.553 | 0.148 | 0.81 | 0.002 |
| post-kASR | CAPT score | 0.982 | −1.713 | 0.314 | 0.74 | 0.007 |

The columns *x*, *y*, *a*, *b*, *S.E.*, and *r* of Table 7 refer to the dependent variable, independent variable, slope of the line, intercept of the line, standard error, and Pearson coefficient, respectively. The first row of Table 7 and the left graph of Figure 3 represent the moderate positive Pearson correlation found between the gASR and kASR pre-test scores ($r = 0.51$, $p = 0.005$), whereas the second row of Table 7 and the right graph of Figure 3 show the moderate positive Pearson correlation found between the gASR and kASR post-test scores ($r = 0.57$, $p = 0.002$).

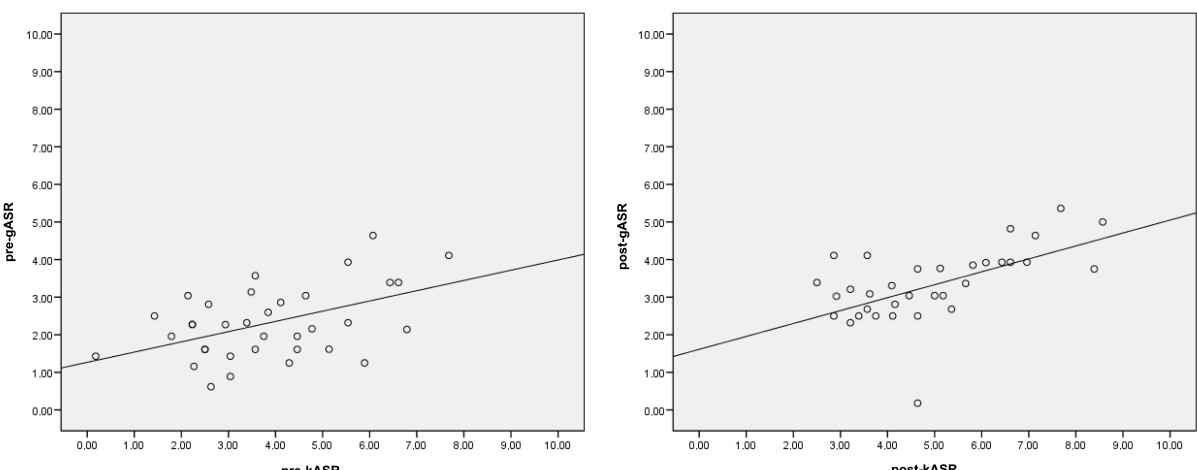

**Figure 3.** Correlation between the gASR and kASR scores of the pre-test (**left** graph) and post-test (**right** graph).

The third row of Table 7 and the left graph of Figure 4 represent the fairly strong positive Pearson correlation found between the CAPT scores and the post-test scores of gASR ($r = 0.81$, $p = 0.002$), whereas the final row of Table 7 and the right graph of Figure 4 show the fairly strong positive Pearson correlation found between the CAPT scores and the post-test scores of the kASR system ($r = 0.74$, $p = 0.007$).

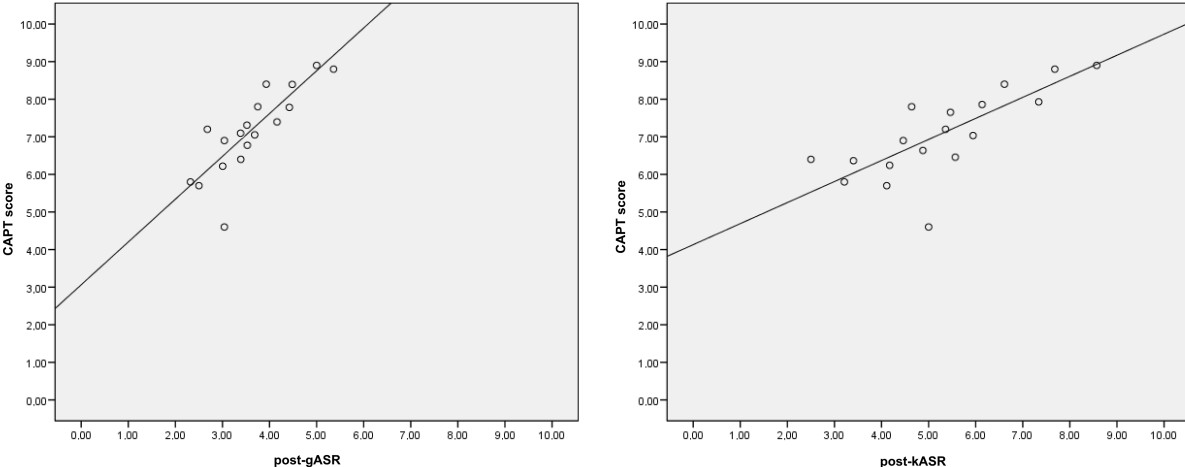

**Figure 4.** Correlation between the gASR (**left** graph) and kASR (**right** graph) ASR scores of the post-test with the CAPT score.

## 4. Discussion

Results showed that the Japañol CAPT tool led experimental group users to carry out a significantly large number of listening, perception, and pronunciation exercises (Table 1). With an effective and objectively registered 57% of the total time, per participant, devoted to training (102.2 min out of 180), high training intensity was confirmed in the experimental group. Each one of the subjects in the CAPT group listened to an average of 612.5 synthesized utterances and produced an average of 291.4 word utterances, which were immediately identified, triggering, when needed, automatic feedback. This intensity of training (hardly obtainable within a conventional classroom) implied a significant level of time investment in tasks, which might establish a relevant factor in explaining the larger gain mediated by Japañol.

Results also suggested that discrimination and production skills were asymmetrically interrelated. Subjects were usually better at discrimination than production (8.5 vs. 10.1 tries per user, see Table 1, #Tries row). Participants consistently resorted to the TTS when faced with difficulties both in perception and production modes (Table 1, #Req.List. row; and Tables 2 and 3, #Lis column). While a good production level seemed to be preceded by a good performance in discrimination, a good perception attempt did not guarantee an equally good production. Thus, the system was sensitive to the expected difficulty of each type of task.

Tables 2 and 3 identified the most difficult phonemes for users while training with Japañol. Users encountered more difficulties in activities related to production. In particular, Japanese learners of Spanish have difficulty with [f] in the onset cluster position in both perception (Table 2) and production (Table 3) [27]. [s]–[θ] present similar results: speakers tended to substitute [θ] by [s], but this pronunciation is accepted in Latin American Spanish [28]. Japanese speakers are also more successful at phonetically producing [l] and [ɾ] than discriminating these phonemes [29]. Japanese speakers have already acquired these sounds since they are allophones of the same liquid phoneme in Japanese. For this reason, it does not seem to be necessary to distinguish them in Japanese (unlike in Spanish).

Regarding the pre/post-test results, we have reported on empirical evidence about the significant pronunciation improvement at the segmental level of the native Japanese beginner-level speakers of Spanish after training with the Japañol CAPT tool (Table 5). In particular, we used two state-of-the-art ASR systems to assess the pre/post-test values. The experimental and in-classroom group speakers improved 0.7|1.1 and 0.6|0.9 points out of 10, assessed by gASR|kASR systems, respectively, after just three one-hour training sessions. These results agreed with previous works which follow a similar methodology [8,30]. Thus, the training protocol and the technology included, such as the CAPT tool and the ASR systems, provided a very useful and didactic instrument that can be used complementary

with other forms of second language acquisition in larger and more ambitious language learning projects.

Our specific-purpose kASR system allowed us to reliably measure the pronunciation quality of the substantial quantity of utterances recorded after testing different training models (Table 4). In particular, this ASR system proved to be useful for working at the segmental (phone) level for non-native speakers. We followed standard preparation procedures for the models, restricted to closed small vocabulary tasks, where the words were selected according to minimal pairs trained with our tool. In this way, what is novel is the fact that we started from a native pronunciation model and transferred it directly to Japanese speakers. Developing an in-house ASR system allowed us not only to customize the post-analysis of the speech without the black-box and pricing limitations of the general-purpose gASR system, but also to neither pre-discard specific words (e.g., out-of-context, infrequent, and very short words) nor worry about the data privacy and preparation costs. Moreover, future research studies might follow the same procedure to develop a similar ASR system for minimal pairs focusing on specific sounds. Despite the positive results reported about the kASR, the training corpus was limited in both quantity and variety of words and the experiment was carried out under a controlled environment. Data augmentation, noise reduction, and a systematic study of the non-native speech data gathered to find pronunciation mistakes associated with key features of proficiency-level characterization with the help of experts for its automatic characterization [4,17] must be considered in the future to expand the project.

Finally, we compared the scores provided by kASR to the gASR ones, obtaining moderate positive correlations between them (Table 7 and Figure 3). The post-test values of both gASR and kASR systems also strongly correlated with the final scores provided by the CAPT tool of the experimental group speakers (Table 7 and Figure 4). In other words, although the training words in Japañol were not the same as the pre/post-test ones, the phonemes trained were actually the same and the speakers were able to assimilate the lessons learned from the training sessions to the final post-test. Therefore, we were able to ensure that both scoring alternatives are valid and can be used for assessing Spanish minimal pairs for certain phonemes and contexts (e.g., availability of resources, learning, place, data privacy, or costs), even though our specific-purpose ASR system is not as accurate as gASR (30.0% vs. 44.22% WER values, Table 4). Future work will consist of fine-tuning our kASR system with more speech data and retraining techniques, such as deep or recurrent neural networks, combining both native and non-native speech in order to improve the current results and to obtain a better customization of the ASR system to the specific phoneme-level tasks. Thus, researchers, scholars, and developers can decide which one to integrate into their CAPT tools depending on the tasks and resources available.

## 5. Conclusions

The Japañol CAPT tool allows L1 Japanese students to practice Spanish pronunciation of certain pairs of phonemes, achieving improvements comparable with the ones obtained in in-classroom activities. The use of minimal pairs permits us to objectively identify the most difficult phonemes to be pronounced by initial-level students of Spanish. Thus, we believe it is worth taking into account when thinking about possible teaching complements since it promotes a high level of training intensity and a corresponding increase in learning.

We have presented the development of a specific-purpose ASR system that is specialized in the recognition of single words of Spanish minimal pairs. Results show that the performance of this new ASR system is comparable with that obtained with the general ASR gASR system. The advantage is not only that the new ASR permits substitution of the commercial system, but also that it will permit us in future applications to obtain information about the pronunciation quality at the level of phoneme.

We have seen that ASR systems can help in the costly intervention of human teachers in the evaluation of L2 learners' pronunciation in pre/post-tests. It is our future challenge

to provide information about the personal and recurrent mistakes of speakers that occur at the phoneme level while training.

**Author Contributions:** The individual contributions are provided as follows. Contributions: Conceptualization, V.C.-P., D.E.-M. and C.T.-G.; methodology, V.C.-P., D.E.-M. and C.T.-G.; software, C.T.-G. and V.C.-P.; validation, C.T.-G. and D.E.-M.; formal analysis, V.C.-P. and D.E.-M.; investigation, C.T.-G. , V.C.-P. and D.E.-M.; resources, V.C.-P. and D.E.-M.; data curation, C.T.-G. and V.C.-P.; writing—original draft preparation, C.T.-G.; writing—review and editing, V.C.-P. and D.E.-M.; visualization, C.T.-G. and V.C.-P.; supervision, D.E.-M.; project administration, V.C.-P.; funding acquisition, V.C.-P. and D.E.-M.. All authors have read and agreed to the published version of the manuscript.

**Funding:** This work has been supported in part by the Ministerio de Economía y Competitividad and the European Regional Development Fund FEDER under Grant TIN2014-59852-R and TIN2017-88858-C2-1-R and by the Consejería de Educación de la Junta de Castilla y León under Grant VA050G18.

**Institutional Review Board Statement:** Not applicable.

**Informed Consent Statement:** Informed consent was obtained from all subjects involved in the study.

**Data Availability Statement:** The data presented and used in this study are available on request from the second author by email (eca-simm@infor.uva.es), as the researcher responsible for the acquisition campaign. The data are not publicly available because the use of data is restricted to research activities, as agreed in the informed consent form signed by participants.

**Acknowledgments:** Authors gratefully acknowledge the active collaboration of Takuya Kimura for his valuable contribution to the organization of the recordings at University of Seisen.

**Conflicts of Interest:** The authors declare no conflict of interest.

## Appendix A. Word List for Pre-Test and Post-Test

**Table A1.** Pre-test and post-test word list.

| | | Spanish | | |
|---|---|---|---|---|
| 1 | **caza** | /ˈkaθa/ | **casa** | /ˈkasa/ |
| 2 | **cocer** | /koˈθer/ | **coser** | /koˈser/ |
| 3 | **cenado** | /θeˈnaðo/ | **senado** | /seˈnaðo/ |
| 4 | **vez** | /beθ/ | **ves** | /bes/ |
| 5 | **zumo** | /ˈθumo/ | **fumo** | /ˈfumo/ |
| 6 | **moza** | /ˈmoθa/ | **mofa** | /ˈmofa/ |
| 7 | **cinta** | /ˈθiNta/ | **finta** | /ˈfinta/ |
| 8 | **concesión** | /koNθeˈsioN/ | **confesión** | /koNfeˈsioN/ |
| 9 | **fugo** | /ˈfuɣo/ | **jugo** | /ˈxuɣo/ |
| 10 | **fuego** | /ˈfweɣo/ | **juego** | /ˈxweɣo/ |
| 11 | **fugar** | /fuˈɣar/ | **jugar** | /xuˈɣar/ |
| 12 | **afuste** | /aˈfuʂte / | **ajuste** | /aˈxuʂte/ |
| 13 | **pelo** | /ˈpelo/ | **pero** | /ˈpeɾo/ |
| 14 | **hola** | /ˈola/ | **hora** | /ˈoɾa/ |
| 15 | **mal** | /malɾ/ | **mar** | /maɾ/ |
| 16 | **animal** | /aniˈmal/ | **animar** | /aniˈmaɾ/ |
| 17 | **hielo** | /ˈʤelo/ | **hierro** | /ˈʤerro/ |
| 18 | **leal** | /leˈal/ | **real** | /rreˈal/ |
| 19 | **loca** | /ˈloka/ | **roca** | /ˈrroka/ |
| 20 | **celada** | /θeˈlaða/ | **cerrada** | /θeˈrraða/ |
| 21 | **pero** | /ˈpeɾo/ | **perro** | /ˈperro/ |
| 22 | **ahora** | /aˈoɾa/ | **ahorra** | /aˈorra/ |
| 23 | **enteró** | /ẽnteˈro/ | **enterró** | /ẽnteˈrro/ |
| 24 | **para** | /ˈpaɾa/ | **parra** | /ˈparra/ |
| 25 | **flotar** | /floˈtaɾ/ | **frotar** | /froˈtaɾ/ |
| 26 | **flanco** | /ˈflaŋko/ | **franco** | /ˈfraŋko/ |
| 27 | **afletar** | /afleˈtaɾ/ | **afretar** | /afreˈtaɾ/ |
| 28 | **flotado** | /flotaˈðo/ | **frotado** | /frotaˈðo/ |

The instructions given to the students in the pre-test and post-test are the following:

- Please read carefully the following list of word pairs (Table A1). Read them from top to bottom and from left to right.
- You can read the word again if you think you have mispronounced it.
- All words are accompanied by their phonetic transcription, in case you find it useful.
- You may read looking at the orthographic expression—*cat*—or at the transcription—/kæt/—but read the orthographic text at least one time.

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

## Short Biography of Authors

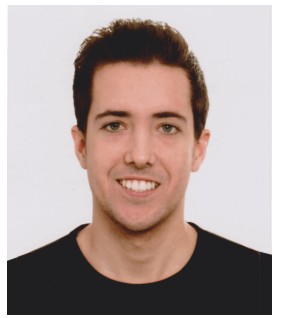

**Cristian Tejedor-García** received the B.A. and the M.Sc. degrees in computer engineering in 2014 and 2016; and the Ph.D. degree in computer science in 2020 from the University of Valladolid, Spain. He is a postdoctoral researcher in automatic speech recognition in the Centre for Language and Speech Technology, Radboud University Nijmegen, Netherlands. He also collaborates in the ECA-SIMM Research Group (University of Valladolid). His research interests include automatic speech recognition, learning games and human-computer-interaction (HCI). He is co-author of several publications in the field of CAPT and ASR.

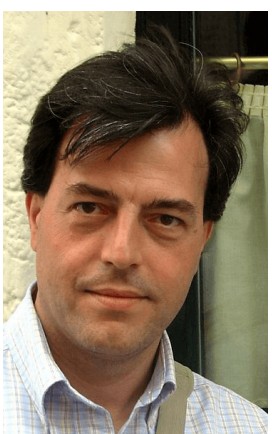

**Valentín Cardeñoso-Payo** received the M.Sc. and the Ph.D. in physics in 1984 and 1988, both from the University of Valladolid, Spain. In 1988, he joined the Department of Computer Science at the same university, where he currently is a full professor in computer languages and information systems. He has been the ECA-SIMM research group director since 1998. His current research interests include machine learning techniques applied to human language technologies, HCI and biometric person recognition. He has been the advisor of ten Ph.D. works in speech synthesis and recognition, on-line signature verification, voice based information retrieval and structured parallelism for high performance computing.

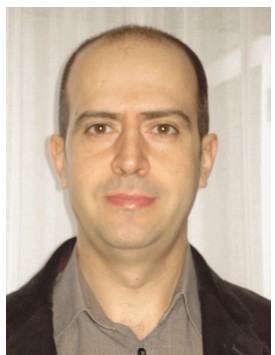

**David Escudero-Mancebo** received the B.A. and the M.Sc. degrees in computer science in 1993 and 1996; and the Ph.D. degree in information technologies in 2002 from the University of Valladolid, Spain. He is an Associate Professor of computer science in the University of Valladolid. He is co-author of several publications in the field of computational prosody (modeling of prosody for TTS systems and labeling of corpora). He has also led several projects on pronunciation training using learning games.