# Peer review of "Automatic Speech Recognition (ASR) Systems Applied to Pronunciation Assessment of L2 Spanish for Japanese Speakers †"

_applsci, doi:10.3390/app11156695_

Round 1

Reviewer 1 Report

The paper entitled "Automatic Speech Recognition (ASR) Systems Applied to Pronunciation Assessment of L2 Spanish for Japanese Speakers"  compare the performance of
author's automatic speech recognition system with Google one.

The overall merit of the presented article is good, but i have some remarks/questions.

  1. The  automatic speech recognition system kASR is not described in the article at all
  2. The novelty of used ASR should be explained more
  3. The comparision with other known ASR should be performed

Reviewer 2 Report

The authors propose an approach for automatic speech recognition (ASR) systems applied to pronunciation assessment of L2 spanish for japanese speakers. 

The paper is well written and structured.

 Here are some comments regarding the paper:-

  1. in line 45, the authors cite the Kaldi ASR Toolkit which provides two speech recognition models, one based on GMM-HMM and the other based on a neural network. my question to the author, why did they prefer to use the GMM-HMM Kaldi model rather than the neural network-based one? .. knowing that the performance of neural network models surpasses the performance of GMM-HMM
  2. Paragraphs from line 364 to 403 of section 3.2. are a confusing cause of inserting a lot of variables / numeric numbers into text. this makes it harder to understand the outcome discussions. I suggest that the authors try to simplify the presentation of the results. maybe they can divide the tables.
